

# Ultrastructure of the Jurassic serpulid tubes–phylogenetic and paleoecological implications

Jakub Słowiński[1], Olev Vinn[2] and Michał Zatoń[1]

[1] Institute of Earth Sciences, University of Silesia, Katowice, Sosnowiec, Poland
[2] Department of Geology, University of Tartu, Tartu, Estonia

## ABSTRACT

The ultrastructural diversity of the Middle and Late Jurassic serpulid tubes from the Polish Basin has been investigated. The inspection of 12 taxa representing the two major serpulid clades allowed for the identification of three ultrastructure types—irregularly oriented prismatic structure (IOP), spherulitic prismatic structure (SPHP), and simple prismatic structure (SP). Six of the studied species are single-layered and six species possess two distinct layers. Ultrastructural diversity corresponds to certain serpulid clades. The members of Filograninae have single-layered tube walls composed of possibly plesiomorphic, irregularly oriented prismatic structure (IOP). Two-layered tubes occur solely within the clade Serpulinae, where the external, denser layer is built of either the ordered spherulitic (SPHP) or simple prismatic microstructure (SP), and the internal layer is composed of irregularly oriented prismatic structure (IOP). Apart from phylogenetic signals provided by the tube ultrastructure, it can be used in analyzing paleoecological aspects of tube-dwelling polychaetes. Compared to the more primitive, irregularly oriented microstructures of Filograninae, the regularly oriented microstructures of Serpulinae need a higher level of biological control over biomineralization. The advent of the dense outer protective layer (DOL) in serpulids, as well as the general increase in ultrastructure diversity, was likely a result of the evolutionary importance of the tubes for serpulids. Such diversity of the tube ultrastructural fabrics allowed for maximizing functionality by utilizing a variety of morphogenetic programs. The biomineralization system of serpulids remains more complex compared to other tube-dwelling polychaetes. Physiologically more expensive tube formation allows for mechanical strengthening of the tube by building robust, strongly ornamented tubes and firm attachment to the substrate. Contrary to sabellids, which perform a fugitive strategy, an increased tube durability allows serpulids a competitive advantage over other encrusters.

**Submitted** 1 March 2024
**Accepted** 23 April 2024
**Published** 23 May 2024

Corresponding author
Jakub Słowiński,
jakub.slowinski@us.edu.pl

# INTRODUCTION

Numerous polychaete families produce habitation tubes (*Rouse & Pleijel, 2001*). Tubicolous polychaetes may either agglutinate exogenous material, such as sand particles and shell fragments, using a proteinaceous cement to form a tube (*e.g., Stewart et al., 2004*; *Zhao*

*et al., 2005*; *Fournier, Etienne & Le Cam, 2010*; *Vinn & Luque, 2013*) or produce secretions by themselves utilizing a variety of glands (*e.g.*, *Hausen, 2005*; *Tanur et al., 2010*). Secreted tubes may be composed of organic substances, such as proteins and polysaccharides (*e.g.*, *Barnes, 1965*; *Chamoy et al., 2001*; *Nishi & Rouse, 2013*), and mineral substances such as calcium carbonate (*Weedon, 1994*; *Fischer, Pernet & Reitner, 2000*; *Vinn et al., 2008*). Amongst tube-dwelling polychaetes producing hard, mineralized exoskeletons are the families Sabellidae, Cirratulidae, and Serpulidae. Whereas calcareous sabellids (*Perkins, 1991*; *Vinn, ten Hove & Mutvei, 2008*; *Słowiński, Banasik & Vinn, 2023*) and cirratulids (*Reish, 1952*; *Fischer, Oliver & Reitner, 1989*; *Fischer, Pernet & Reitner, 2000*; *Taylor et al., 2010*; *Kočí et al., 2021*; *Guido et al., 2024*) are restricted to one (*Glomerula*) and two genera (*Dodecaceria*, *Diplochaetetes*) in each family respectively, only serpulids dwell exclusively in tubes composed of calcium carbonate (*Vinn et al., 2008*).

Thus, having a hard, mineralized exoskeleton, serpulids are the most abundant polychaetes in the fossil record, appearing as far back as the Middle Permian (*Sanfilippo et al., 2017*; *Sanfilippo et al., 2018*). However, their hard parts are not strictly related to the organism's soft body and form only a habitation tube serving, among others, for protection against predators (*e.g.*, *Morton & Harper, 2009*; *Klompmaker, 2012*). Although tubes' taxonomic usefulness is not without validity, as many genera possess easily recognizable tubes, which are diagnostic displaying different longitudinal and transverse elements (see *e.g.*, *Ippolitov et al., 2014*), homeomorphy is still a common feature in serpulid tube morphology (*e.g.*, *Kupriyanova & Ippolitov, 2015*) due to convergence and high ecophenotypic plasticity. This leads to certain discrepancies between paleontological and biological classification, the latter of which is based solely on the molecular data, soft parts, and their reciprocal relationships (*Kupriyanova, Macdonald & Rouse, 2006*; *ten Hove & Kupriyanova, 2009*). Moreover, it makes the linkage of contemporary species with their ancestors difficult due to the lack of fully reliable taxonomic tools.

Serpulid tubes can be composed of up to four layers; however, most of them are single-layered (*Vinn et al., 2008*). They exhibit different ultrastructural fabrics, depending on how calcium carbonate crystals are arranged and oriented (see *Vinn et al., 2008*). During the progressing expansion of the tube, an animal secretes consecutive growth lamellae, which correspond to a single growth episode. Such growth lines may be either straight, perpendicular to the direction of growth (*e.g.*, *Bałuk & Radwański, 1997*) or much more frequently chevron-shaped (see *Weedon, 1994*). Serpulid ultrastructures may provide phylogenetic signals in the case of Jurassic taxa but also serve as a record of the physiological changes of the worm during its entire lifespan, providing important ecological signatures that may be successfully implemented in paleoecological analyses. Moreover, serpulid tubes may exhibit additional skeletal structures such as tubulae, alveolar structures, or internal tube structures, helping in unraveling their true systematics (*e.g.*, *Thomas, 1940*; *Pillai, 1993*; *Pillai & ten Hove, 1994*; *Jäger, 2005*).

Recent studies have proven a tube microstructure to be a useful tool in deciphering the true affinity of fossil serpulid (*e.g.*, *Bornhold & Milliman, 1973*; *Vinn, ten Hove & Mutvei, 2008*; *Vinn & Kupriyanova, 2011*; *Kupriyanova & Ippolitov, 2012*; *Vinn, 2013*; *Buckman, 2020*; *Buckman & Harries, 2020*), and other calcareous tube-dwelling polychaete species

(*Vinn, ten Hove & Mutvei, 2008*; *Taylor et al., 2010*; *Słowiński, Banasik & Vinn, 2023*). However, the majority of them were conducted based on single species, and/or single specimens.

In the present study, we performed a microstructural investigation of the Jurassic serpulid tubes based on the representatives of the two main serpulid clades according to the newest systematics (*Kupriyanova, ten Hove & Rouse, 2023*). We provide paleoecological implications of different microstructures and consequently biomineralization system and highlight some phylogenetic signals and evolutionary patterns of distinct microstructures.

## MATERIAL AND METHODS

The examined fossils consist of 47 specimens of serpulids representing 12 taxa (Table 1). The studied material has been selected from an ample collection (see *Słowiński et al., 2020*; *Słowiński et al., 2022*) concerning the well-preserved specimens, which allowed for a reliable ultrastructural investigation. All diagenetically altered tubes have been discarded from further studies. Almost all specimens encrust a variety of invertebrate fossils, oncoids, and hiatus concretions, and have been derived from various localities with Middle and Upper Jurassic (upper Bajocian–lower Kimmeridgian) deposits (Fig. 1) representing an array of paleoenvironments. The investigated material is stored at the Institute of Earth Sciences in Sosnowiec, abbreviated GIUS 8-3589 (Callovian of Zalas), GIUS 8-3730 (middle Bathonian of Gnaszyn Dolny), GIUS 8-3745 (upper Bathonian-lower Callovian of Bolęcin), GIUS 8-3746 (Oxfordian of Zalas), GIUS 8-3747 (lower Kimmeridgian of Małogoszcz), GIUS 8-3750 (Bajocian-Bathonian of Ogrodzieniec-Świertowiec), GIUS 8-3751 (upper Bathonian of Krzyworzeka and upper Bathonian of Żarki). Their paleogeographic and geological background is beyond the scope of the present research and has already been discussed in detail by *Słowiński et al. (2020)* and *Słowiński et al. (2022)*.

Before the preparation of specimens for the SEM examination, they were coated with ammonium chloride and photographed using the Canon EOS 350D digital camera. All previously selected samples were cut longitudinally, polished, and etched with a 5% solution of acetic acid for one minute prior to the SEM study. Part of these tube portions were oriented and mounted in epoxy resin before polishing. Subsequently, the serpulid tube microstructure investigations were performed on a SEM Zeiss EVO MA15 under a back-scattered electron detector (BSE) in a low vacuum regime (Department of Geology, University of Tartu, Estonia). The sections were also inspected under an environmental scanning electron microscope (ESEM) Philips XL30, and a Quanta 250 scanning electron microscope using BSE imaging (both at the Institute of Earth Sciences in Sosnowiec). The beam voltage was operated at 20 kV.

## RESULTS

During the examination of 12 taxa within eight genera representing two major clades, three types of ultrastructure were distinguished: irregularly oriented prismatic structure (IOP), spherulitic prismatic structure (SPHP), and simple prismatic structure (SP) (Table 1).

**Table 1  Ultrastructural diversity of the Jurassic serpulid tubes from Poland.**

| Taxon | Ultrastructure | | | Age | Locality | Number of individuals investigated |
|---|---|---|---|---|---|---|
| | IOP | SPHP | SP | | | |
| *Metavermilia* cf. *striatissima* * | 1 | | | upper Bathonian | Krzyworzeka | 3 |
| *Filogranula runcinata* * | [2] | 1 | | middle Bathonian, Callovian | Gnaszyn Dolny, Zalas | 5 |
| *Filogranula spongiophila* * | | | 1 | Oxfordian | Zalas | 3 |
| *Cementula spirolinites* ● | | | 1 | Oxfordian | Zalas | 9 |
| *Cementula radwanskae* ● | | | [1] | Callovian | Zalas | 4 |
| *Propomatoceros lumbricalis* × | 2 | 1, [3] | | Bajocian-Bathonian, middle Bathonian, upper Bathonian, Bathonian-Callovian, Callovian, lower Kimmeridgian | Ogrodzieniec-Świertowiec, Gnaszyn Dolny, Żarki, Bolęcin, Zalas, Małogoszcz | 8 |
| *Nogrobs* aff. *quadrilatera* × | 2 | | 1 | middle Bathonian | Gnaszyn Dolny | 3 |
| *Nogrobs*? aff. *tricristata* × | 2 | | 1 | middle Bathonian | Gnaszyn Dolny | 1 |
| *Nogrobs* aff. *tetragona* × | 2 | | 1 | middle Bathonian | Gnaszyn Dolny | 2 |
| *Mucroserpula* sp. × | 2 | 1 | | middle Bathonian | Gnaszyn Dolny | 3 |
| *Placostegus planorbiformis* × | | | 1 | Oxfordian | Zalas | 3 |
| Serpulidae sp. × | 2 | | 1 | middle Bathonian | Gnaszyn Dolny | 3 |
| **Total** | | | | | | 47 |

Notes.
    IOP, irregularly oriented prismatic structure; SPHP, spherulitic prismatic structure; SP, simple prismatic structure.
    Numbers (1, 2, 3) indicate the position of the layer within the tube counted from the exterior. Square brackets denote partially preserved or equivocal microstructure precluding a fully reliable recognition. Asterisk, circle, and ×mark correspond to Filograninae, Serpulini, and Ficopomatini respectively. All the taxa have been described taxonomically in *Słowiński et al. (2022)*.

### Filograninae (BI)

Members of the clade Filograninae, formerly referred to as BI (see *Kupriyanova, Macdonald & Rouse, 2006*; *Kupriyanova et al., 2009*), are represented in the herein material by two genera: *Metavermilia* and *Filogranula*, the latter one consisting of two species: *F. runcinata* (*Sowerby, 1829*) and *F. spongiophila* (*Słowiński et al., 2022*).

The tube wall of *Metavermilia* cf. *striatissima* (*Fürsich, Palmer & Goodyear, 1994*) is single-layered, and composed of an irregularly oriented prismatic structure (IOP) (Fig. 2). Minute (maximally a few μm), needle-like crystals are deployed more or less evenly within the entire tube wall. The longitudinal axes of crystals lack a uniform orientation.

The tube of *Filogranula runcinata* is presumably single-layered and is composed of a spherulitic, regularly oriented prismatic microstructure (SPHP) (Fig. 3). It is formed by crystals of prismatic shape exhibiting a slightly spherulitic arrangement. Some internal parts of the tube are built of irregularly oriented, tiny, elongated crystals indicating an irregularly oriented prismatic structure (IOP). However, the boundary between the two putative layers is transitional and the latter microstructure occurs rather like inclusions in certain areas of the tube wall. Growth lines are apparent across almost the entire tube. Some minor, external parts of the section may appear like fine homogeneous granular microstructure; however, more likely it corresponds to the differences in preservation

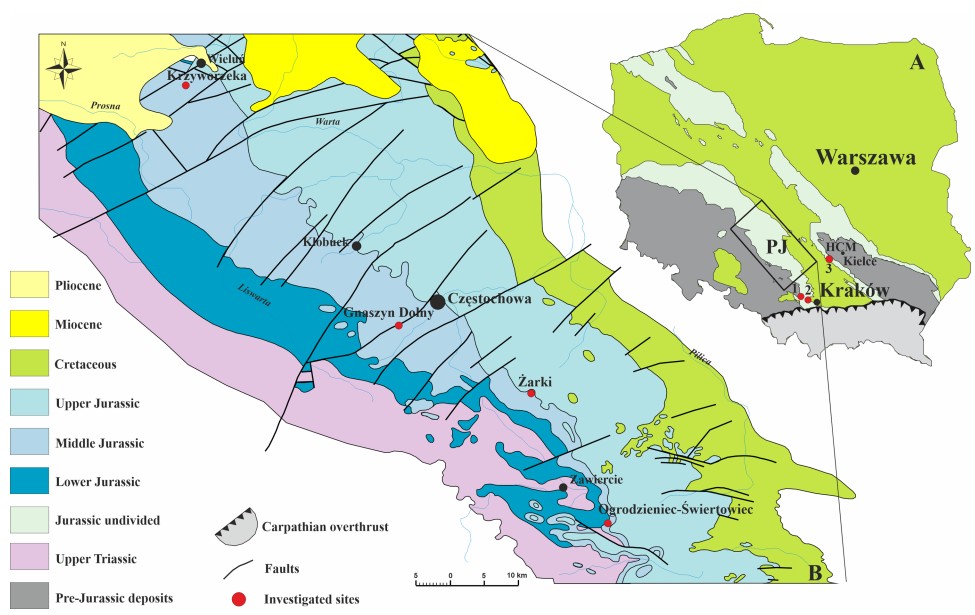

**Figure 1 Geology of the investigated area.** (A) Geological sketch-map of Poland without the Cenozoic cover showing three sampled sites; HCM–Holy Cross Mountains; PJ–Polish Jura; 1–Bolęcin; 2–Zalas; 3–Małogoszcz. (B) Geological map of the Polish Jura area without Quaternary cover, with sampled localities indicated (modified after *Zatoń, Marynowski & Bzowska, 2006*).

between the external and internal parts of the tube due to the diagenetic alteration of irregularly oriented prismatic microstructure.

The tube of *Filogranula spongiophila* consists of a single layer that is composed of a simple, regularly oriented prismatic structure (SP) (Fig. 4). Growth increments are visible across almost the whole tube (Figs. 4C–4E).

## Serpulinae, tribe Serpulini (AI)

This tribe is represented in the investigated material by one genus comprising two species: *Cementula spirolinites* (Münster in *Goldfuss, 1831*) and *Cementula radwanskae* (*Słowiński et al., 2022*).

The tube wall of *C. spirolinites* is single-layered and consists of a simple prismatic structure (SP) (Fig. 5). This microstructure is formed by parallel prismatic crystals oriented perpendicularly to each growth line showing incremental zonation. The crystals are arranged perpendicularly or obliquely to the tube wall.

The tube microstructure of the investigated specimens of *C. radwanskae* is altered in most places. However, some places exhibiting faint growth increments indicate possibly a simple (SP) or spherulitic prismatic structure (SPHP) constituting a single layer (Fig. 6). Certain areas with tiny, elongated irregularly oriented crystals presumably are an effect of a diagenetic distortion of the tube wall.

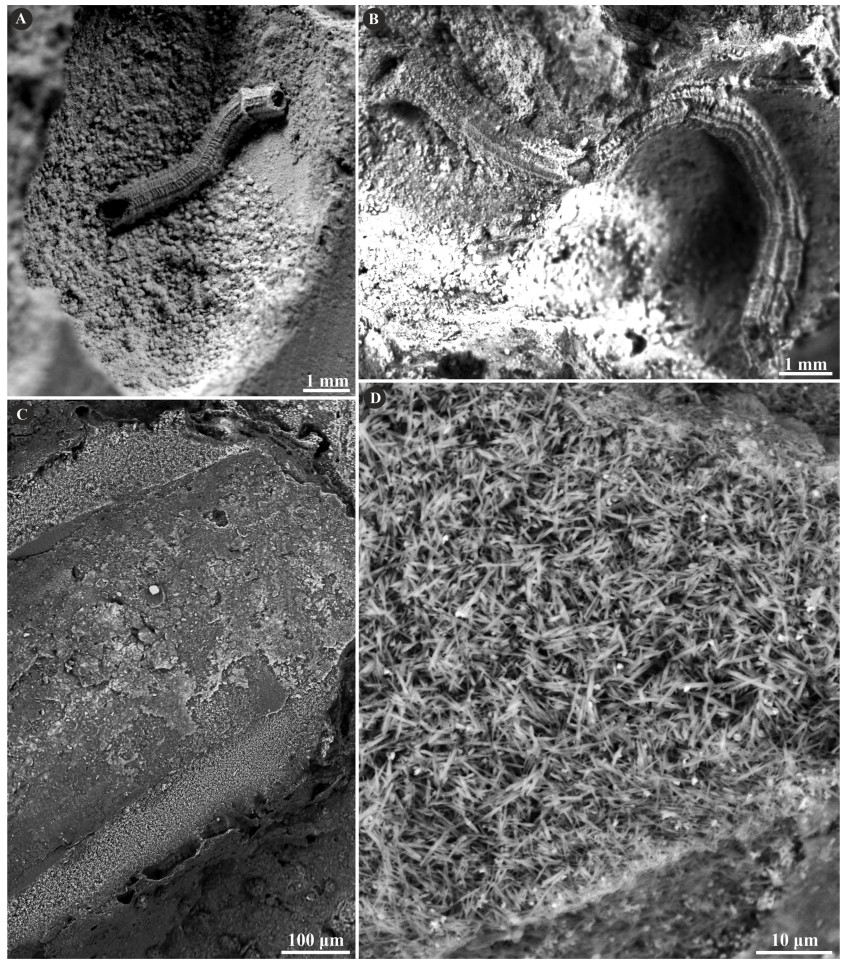

**Figure 2** **Selected specimens of *Metavermilia*. cf. *striatissima* (*Fürsich, Palmer & Goodyear, 1994*) from the Jurassic of Poland.** (A, B) *Metavermilia* cf. *striatissima* encrusting the interior of the boring *Gastrochaenolites* from the upper Bathonian of Krzyworzeka, Polish Jura ((A) GIUS 8-3751/3; (B) GIUS 8-3751/8). (C, D) Longitudinal sections of the tubes from the upper Bathonian of Krzyworzeka with visible irregularly oriented prismatic structure (IOP). (C) General look of the ground, polished and etched tube with visible thin, single-layered walls (GIUS 8-3751/3). (D) Close-up of the tube wall section showing densely packed, tiny, elongated prismatic crystals of irregular arrangement (GIUS 8-3751/3).

### Serpulinae, tribe Ficopomatini (AII)

The tribe Ficopomatini, formerly referred to as AII (see *Kupriyanova, Macdonald & Rouse, 2006*; *Kupriyanova et al., 2009*), includes here five genera with seven species. *Propomatoceros lumbricalis* (*von Schlotheim, 1820*) possesses a tube wall consisting of two or three layers (Fig. 7). The majority of the examined specimens have two-layered tubes. The external part is formed by a spherulitic prismatic structure (SPHP), which consists of parallel crystals of a slightly spherulitic arrangement. The inner part has an irregularly oriented prismatic microstructure (IOP) comprising elongated, but relatively short, tiny crystals with inconsistent orientation axes. In some areas of the tubes' wall, the IOP structure is diagenetically altered superficially resembling a fine homogeneous granular structure. The

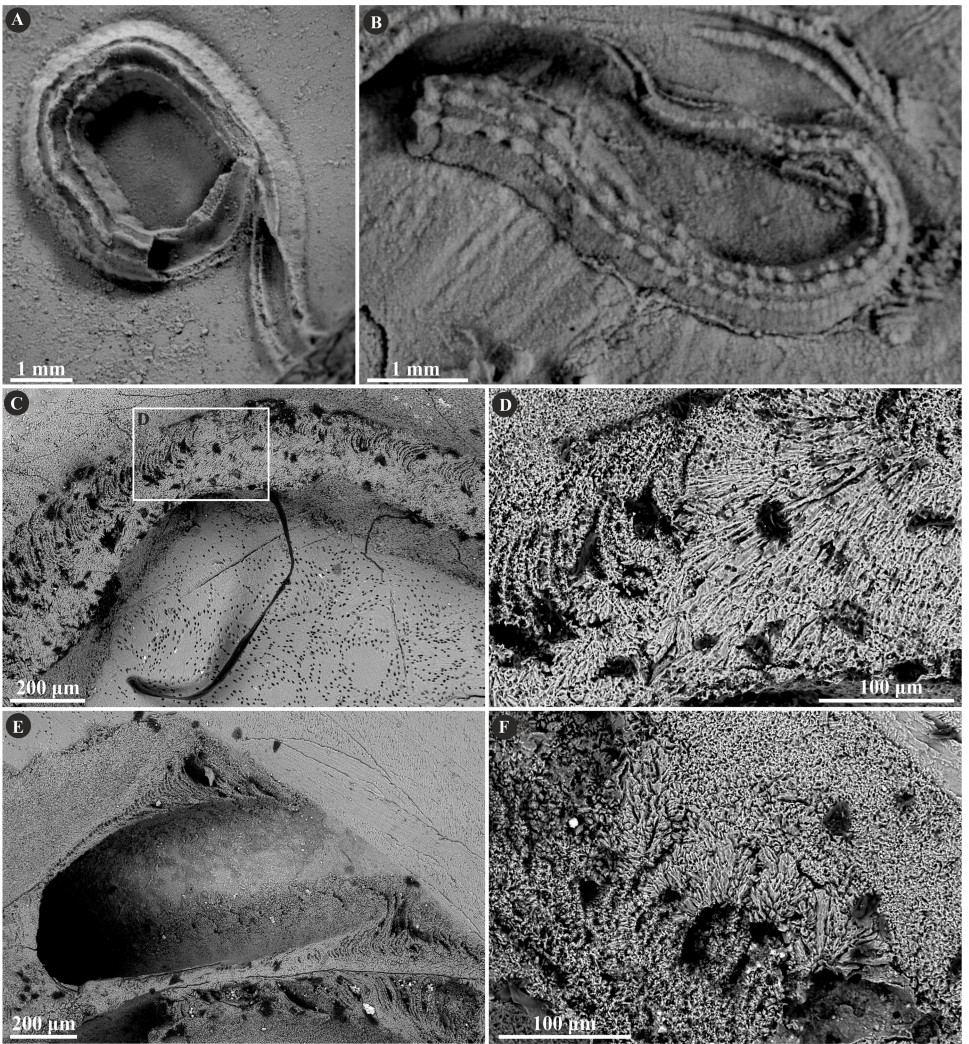

**Figure 3  Selected specimens of *Filogranula runcinata* (*Sowerby, 1829*) from the Jurassic of Poland.** (A, B) *Filogranula runcinata* from the middle Bathonian of Gnaszyn Dolny (A. GIUS 8-3730/10) and the Callovian of Zalas (B. GIUS 8-3589/6) encrusting shell fragments. (C–F) Longitudinal sections of the tubes showing a single-layered tube wall built of spherulitic prismatic structure (SPHP) with well-visible growth increments (GIUS 8-3730/10).

putative three-layered tube wall (Fig. 7D) is built of the external and internal (uncertain) parts composed of a spherulitic prismatic structure (SPHP). The middle part shows a fine homogeneous granular microstructure, which may be either a primary microstructure or similarly to other specimens, an obliterated IOP structure. In contrast to two-layered *P. lumbricalis* tubes, no areas exhibiting irregularly oriented prismatic structures have been found. As such, it is not evident, whether these microstructures are primary or have undergone diagenetic alteration. The external parts exhibit incremental zonation with prominent chevron-shaped growth lines. The crystals are continuously and regularly positioned through successive growth increments.

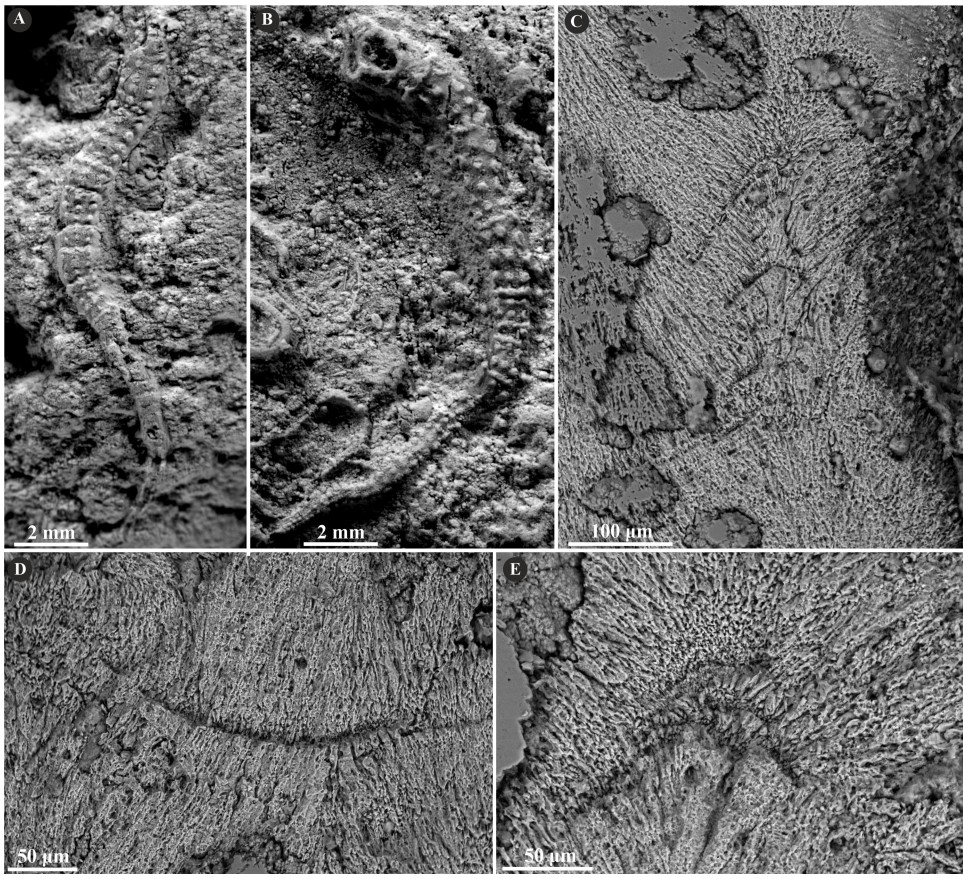

**Figure 4** **Selected specimens of *Filogranula spongiophila* (*Słowiński et al., 2022*) from the Jurassic of Poland.** (A, B) *Filogranula spongiophila* from the Oxfordian of Zalas ((A) GIUS 8-3746/2; (B) GIUS 8-3746/3) encrusting sponge fragments. (C–E) Longitudinal sections of the tubes showing the simple prismatic structure (SP) with well-visible growth increments (GIUS 8-3746/9).

Herein, *Nogrobs* is represented by three species: *N.* aff. *quadrilatera* (*Goldfuss, 1831*), *N.* aff. *tetragona* (*Sowerby, 1829*), and *N*? aff. *tricristata* (*Goldfuss, 1831*), which, however, share all the major microstructural characters. The tube walls of all the species have two layers, which are separated from each other with a sharp boundary. The external layer is composed of a simple prismatic structure (SP), whereas the internal layer is composed of very thin, short, needle-like crystals, which make up an irregularly oriented prismatic structure (IOP) (Fig. 8).

*Mucroserpula* sp. (*Regenhardt, 1961*) possesses a two-layered tube wall (Fig. 9). The internal layer is composed of an irregularly oriented prismatic structure (IOP) consisting of bunches of densely packed, short but elongated crystals lacking uniform orientation axes. The external layer is composed of a regular spherulitic prismatic structure (SPHP), which constitutes the major part of the tube wall. Within this layer, crystals are oriented perpendicularly with respect to each incremental zone and have a somewhat prismatic arrangement. The chevron-shaped growth lines are very well-visible alongside almost the whole tube length. The border between the two layers is somewhat transitional.

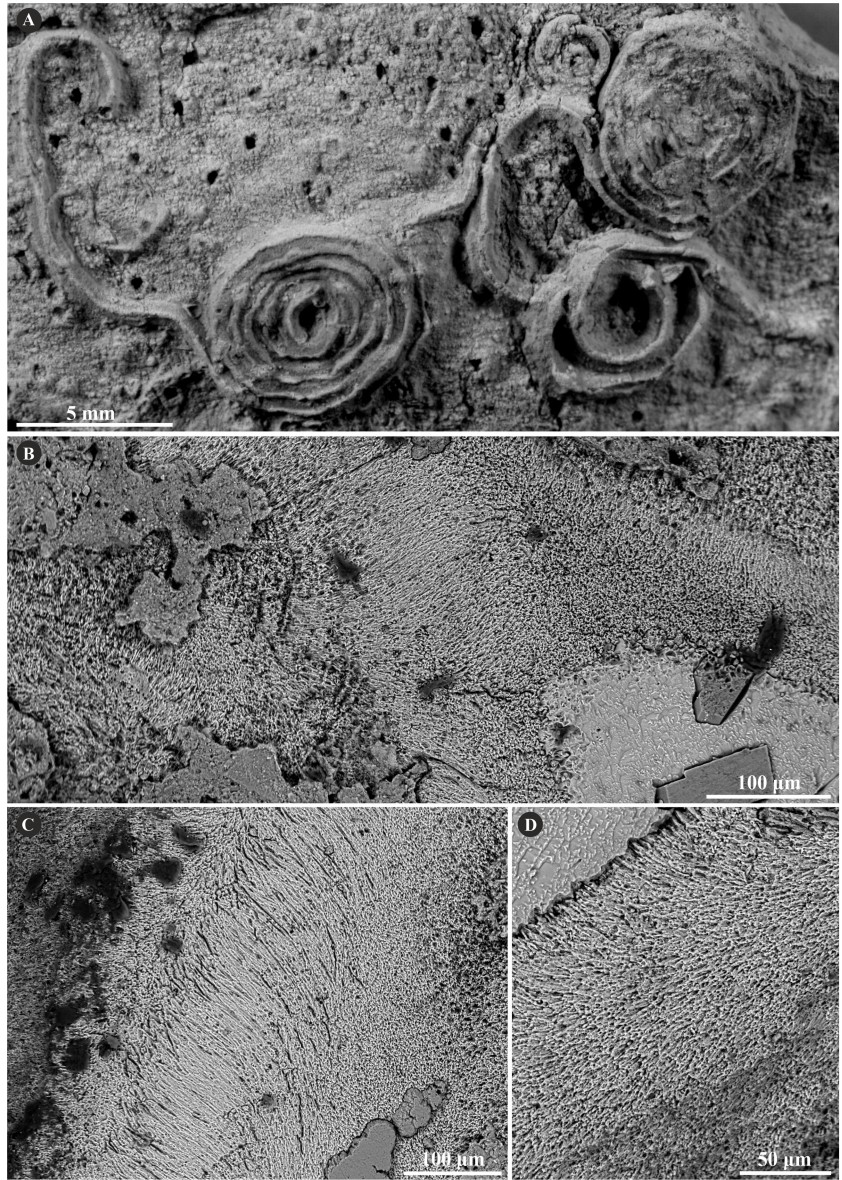

**Figure 5  Selected specimens of *Cementula spirolinites* (Münster in *Goldfuss, 1831*) from the Jurassic of Poland.** (A) *Cementula spirolinites* from the Oxfordian of Zalas encrusting a sponge fragment (GIUS 8-3746/4). (B–D) Longitudinal sections of the tubes showing the simple prismatic structure (SP). (B) GIUS 8-3746/10; (C, D) GIUS 8-3746/11.

The tube wall of *Placostegus planorbiformis* (Münster in *Goldfuss, 1831*) is single-layered, and composed of a simple prismatic structure (SP) (Fig. 10). All crystals are more or less parallel within each incremental zone. Within the outermost part of the tube, the crystals are oriented perpendicularly to the tube wall.

The specimens of undetermined serpulid, Serpulidae sp. possess a two-layered tube wall (Fig. 11). The external part features a regularly oriented simple prismatic structure (SP). The internal, thinner part, separated from the external with a sharp boundary is formed

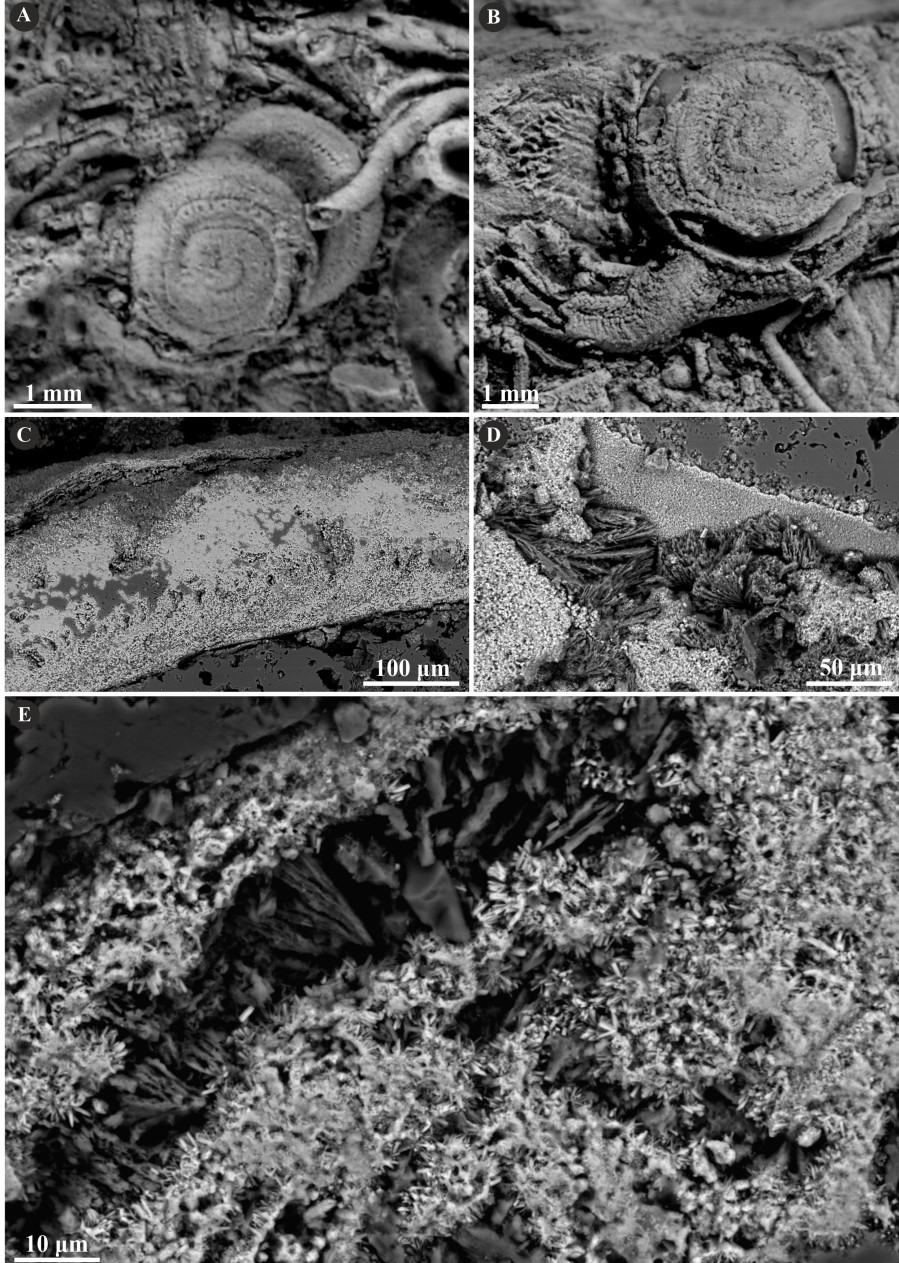

**Figure 6** **Selected specimens of *Cementula radwanskae* (*Słowiński et al., 2022*) from the Jurassic of Poland.** (A, B) *Cementula radwanskae* from the Callovian of Zalas encrusting shell fragments (A) 8-3589/7; (B) GIUS 8-3589/10). (C–E) Longitudinal sections of the tubes showing mostly altered, presumably simple prismatic microstructure (SP); GIUS 8-3589/18. Note the weak growth increments in C.

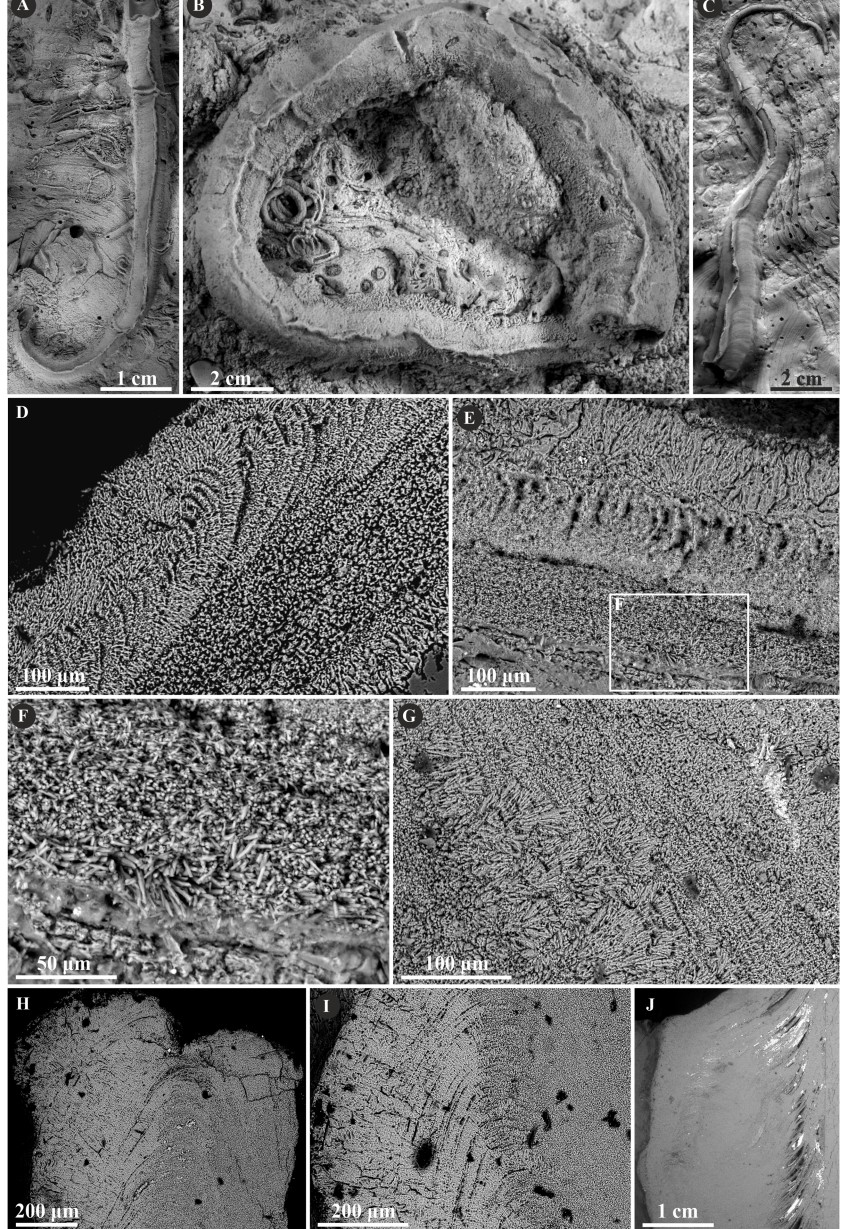

**Figure 7** **Selected specimens of *Propomatoceros lumbricalis* (Schlotheim, 1820) from the Jurassic of Poland.** (A, B) *Propomatoceros lumbricalis* encrusting a shell fragment from the Callovian of Zalas (A) GIUS 8-3589/11; (B) GIUS 8-3589/12). (C) *Propomatoceros lumbricalis* encrusting an oyster shell from the middle Bathonian of Gnaszyn Dolny (GIUS 8-3730/12). (D–J) Sections of the tubes showing the ultra-structural details. (continued on next page…)

**Figure 7 (…continued)**
(D) The longitudinal section of the *P. lumbricalis* tube from the upper Bathonian–lower Callovian of Bolęcin showing the putative three-layered tube (GIUS 8-3745/5). The external layer shows spherulitic prismatic structure (SPHP) with growth increments; the middle part exhibits presumably altered irregularly oriented prismatic structure (IOP) superficially resembling homogeneous structure; the internal layer shows uncertain spherulitic prismatic structure. (E) The longitudinal section of the two-layered *P. lumbricalis* tube from the middle Bathonian of Gnaszyn Dolny showing the external spherulitic prismatic structure (SPHP) with growth increments and the internal irregularly oriented prismatic structure (IOP) (GIUS 8-3730/30). (F) A close-up showing the internal layer built of irregularly oriented prismatic structure (IOP) with short, needle-like crystals of different length and orientation (GIUS 8-3730/30). (G) The longitudinal section of the *P. lumbricalis* tube from the lower Kimmeridgian of Małogoszcz (GIUS 8-3747/5) showing the spherulitic arrangement of crystals within the external tube layer. (H, I) Longitudinal sections of the two *P. lumbricalis* tubes from the middle Bathonian of Gnaszyn Dolny showing the external spherulitic prismatic structure (SPHP) with well-visible growth increments and the internal, partially recrystallized irregularly oriented prismatic structure (IOP) (H. GIUS 8-3730/31; I. GIUS 8-3730/32). (J) The lateral section of the *P. lumbricalis* tube from the middle Bathonian of Gnaszyn Dolny showing tubulae subdivided densely by septae (GIUS 8-3730/33).

by an irregularly oriented prismatic structure (IOP) composed of densely packed, minute, elongated crystals.

## DISCUSSION

### Tube ultrastructure evolution and its phylogenetic constraints

Three distinct types of ultrastructure have been identified within 12 taxa corresponding to the two of three main clades of serpulids—Filograninae and Serpulinae (*Kupriyanova, ten Hove & Rouse, 2023*). These microstructures comprise irregularly oriented prismatic structure (IOP), spherulitic prismatic structure (SPHP), and simple prismatic structure (SP) (see Table 1), which are among the most prevalent microstructure type in fossil serpulids (*Vinn et al., 2008*; *Vinn, 2020*). Six of these taxa have single-layered tubes and six are two-layered, of which one taxon perhaps may possess either two or three ultrastructural layers (see discussion). The majority of both Cenozoic and contemporary serpulids are single-layered, only about one-third of serpulid species have at least two or up to four distinct ultrastructural layers (*Vinn et al., 2008*). During the Jurassic, the percentage of multi-layered serpulid species was lower, constituting approximately 25% (*Vinn & Furrer, 2008*). It may have resulted from the evolutionary pattern where more complex tube walls with at least two layers were more common from the beginning of the Cenozoic onward— the growing complexity of microstructures combined with relatively fast biomineralization likely enhanced the strengthening abilities of the tubes. It can also be explained by the vagaries of the fossil record resulting in incomplete preservation of the outermost layers, which might be similar to the case of calcareous sabellids (*Vinn, ten Hove & Mutvei, 2008*; *Słowiński, Banasik & Vinn, 2023*).

*Sanfilippo (1998)* presented a concept to utilize the ultrastructural diversity of serpulids to assess their generic determination. Further studies revealed that microstructure characters may be species-dependent (*Vinn, 2007*; *Vinn et al., 2008*; *Kupriyanova & Ippolitov, 2015*), which limits the application of tube microstructures in deciphering the generic affiliation of serpulids. Regardless, ultrastructural fabrics of tubes may still be used to approach the

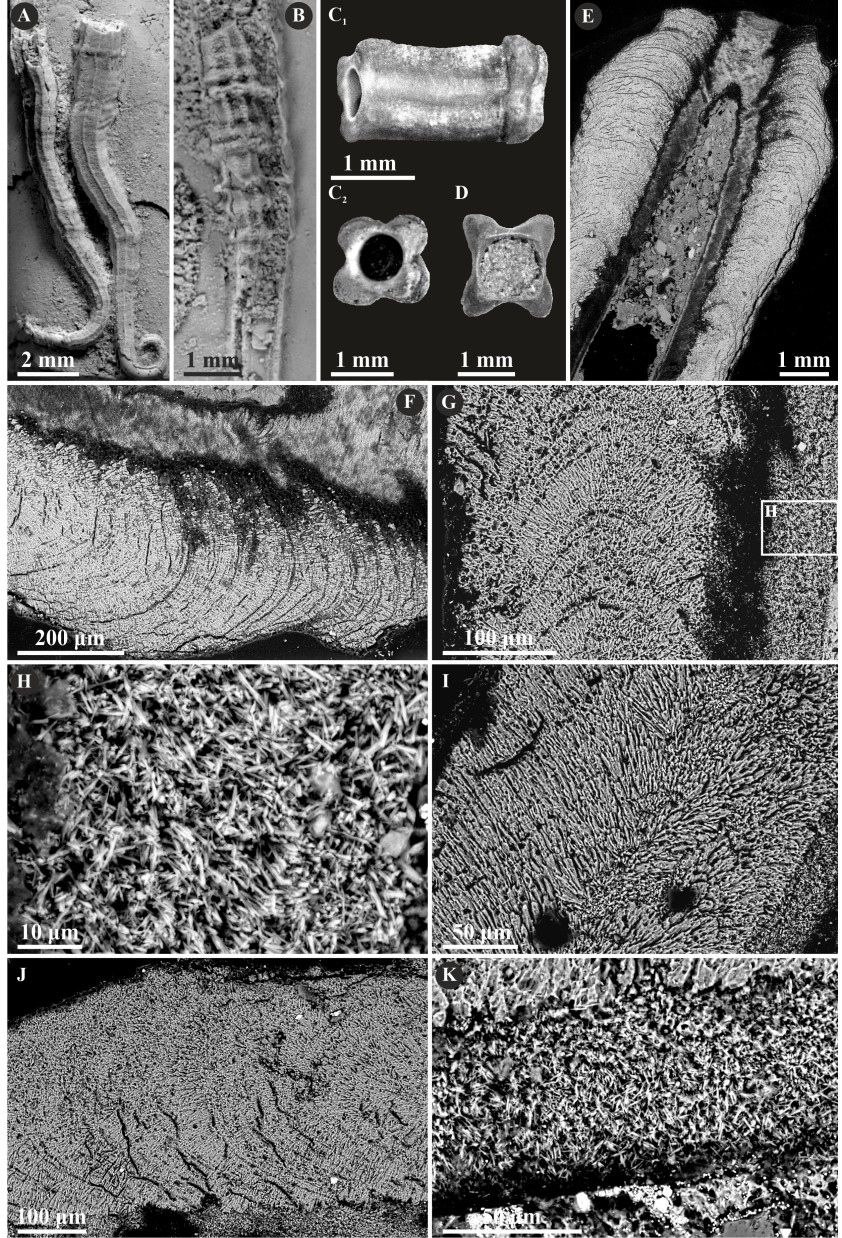

**Figure 8 Selected specimens of *Nogrobs* from the Jurassic of Poland.** (A) Two specimens of *Nogrobs* aff. *quadrilatera* (*Goldfuss, 1831*) encrusting a belemnite rostrum from the middle Bathonian of Gnaszyn Dolny (GIUS 8-3730/17). (B) *Nogrobs*? aff. *tricristata* (*Goldfuss, 1831*) encrusting a belemnite rostrum from the middle Bathonian of Gnaszyn Dolny (GIUS 8-3730/19). (C, D) Free-lying tubes of *Nogrobs* aff. *tetragona* (*Sowerby, 1829*) (C. GIUS 8-3730/24; D. GIUS 8-3730/25); lateral (C₁) and cross-section (C₂, D) view. 

**Figure 8 (…continued)**
(E, F, G, I, J) The longitudinal sections of the *Nogrobs* aff. *tetragona* (E, F. GIUS 8-3730/34), *Nogrobs*? aff. *tricristata* (G. GIUS 8-3730/20), and *Nogrobs* aff. *quadrilatera* (I. GIUS 8-3730/35; J. GIUS 8-3730/36) tubes from the middle Bathonian of Gnaszyn Dolny showing two distinct layers: the external simple prismatic structure (SP) with distinct growth lines and internal irregularly oriented prismatic structure (IOP). (H, K) A close-up of the longitudinal sections of the internal tube layer of *Nogrobs*? aff. *tricristata* (H. GIUS 8-3730/20) and *Nogrobs* aff. *quadrilatera* (K. GIUS 8-3730/36) showing irregularly oriented prismatic structure (IOP) with short, needle-like crystals of different length and orientation.

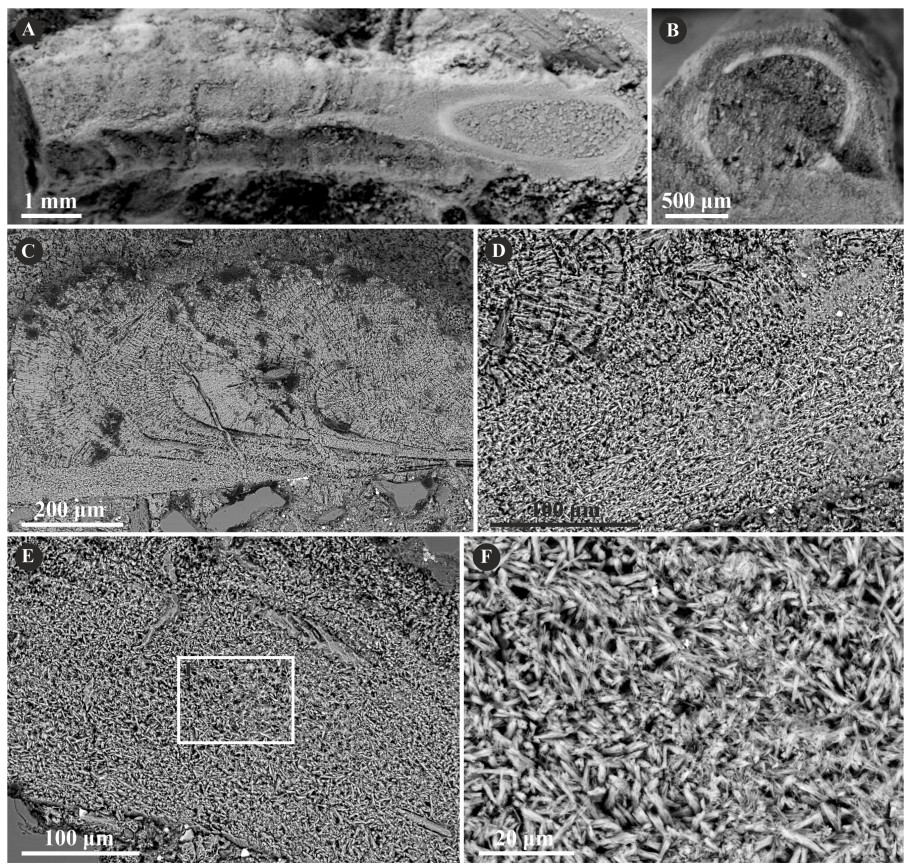

**Figure 9   Selected specimens of *Mucroserpula* (*Regenhardt, 1961*) from the Jurassic of Poland.** (A, B) *Mucroserpula* sp. encrusting a fragment of a shell from the middle Bathonian of Gnaszyn Dolny (GIUS 8-3730/26); top (A) and cross-section view (B). (C–E) Longitudinal sections of the *Mucroserpula* sp. tube from the middle Bathonian of Gnaszyn Dolny showing two distinct layers: the external spherulitic prismatic structure (SPHP) with distinct, chevron-shaped growth lines (C) and internal irregularly oriented prismatic structure (IOP) (GIUS 8-3730/26). (F) A close-up of the longitudinal section of the internal tube layer showing irregularly oriented prismatic structure (IOP) with short, needle-like crystals of different length and orientation (GIUS 8-3730/26).

relationships between distinct types of structures throughout the evolution of the main serpulid clades (*e.g.*, *Vinn et al., 2008*; *Vinn & Kupriyanova, 2011*; *Vinn, 2013*; *Ippolitov & Rzhavsky, 2014*; *Ippolitov & Rzhavsky, 2015a*; *Ippolitov & Rzhavsky, 2015b*).

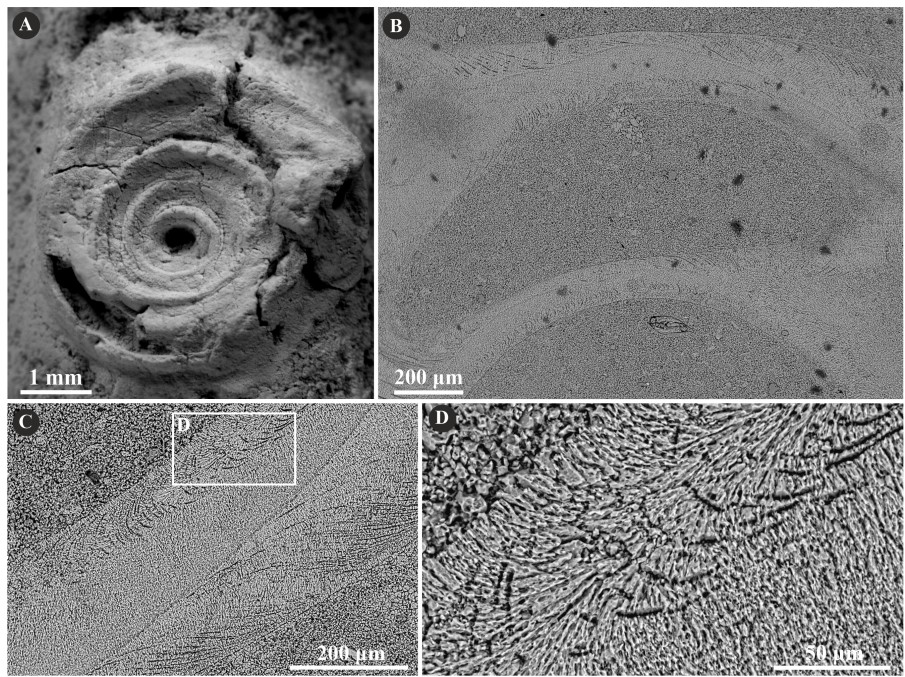

**Figure 10** **Selected specimens of *Placostegus planorbiformis* (Münster in *Goldfuss, 1831*) from the Jurassic of Poland.** (A) *P. planorbiformis* encrusting a sponge fragment from the Oxfordian of Zalas (GIUS 8-3746/8). (B–D) Longitudinal sections of the tubes showing the simple prismatic structure (SP) with well-visible growth increments (GIUS 8-3746/8).

The most recent phylogenetic analyses indicate that serpulids are split into three major clades—Filograninae, Spirorbinae, and Serpulinae, the last of which is further subdivided into two tribes: Serpulini and Ficopomatini (*Kupriyanova, ten Hove & Rouse, 2023*). The family Serpulidae until recently was maintained to comprise two major clades referred to as B and A, separated into BI, BII, AI, and AII (*Kupriyanova, Macdonald & Rouse, 2006*; *Kupriyanova et al., 2009*; *Kupriyanova & Nishi, 2010*). Formerly recognized clades BI and BII generally may be related to Filograninae and Spirorbinae, whereas AI and AII, both settled within Serpulinae, correspond to the tribes Serpulini and Ficopomatini, respectively (*Kupriyanova, Macdonald & Rouse, 2006*; *Kupriyanova et al., 2009*; *Kupriyanova, ten Hove & Rouse, 2023*). Apart from Spirorbinae (former BII), members of all clades are present in the material studied.

Clade Filograninae is possibly rooted even in the Permian with primitive tubes such as *Filograna* (*Sanfilippo et al., 2017*; *Ramsdale, 2021*), and abundantly represented in the fossil record during the Mesozoic by many strongly ornamented tubes with several keels, *e.g.*, *Vermiliopsis*, *Metavermilia*, and *Filogranula*, the latter two of which are present in the material investigated. *Metavemilia* cf. *striatissima* is single-layered and possesses irregularly oriented prismatic microstructure (IOP), which has been also found in the recent *Metavermilia multicristata* (*Vinn et al., 2008*). It is characteristic of this clade and is also the most common microstructure in the recent serpulids encompassing ca 60% of

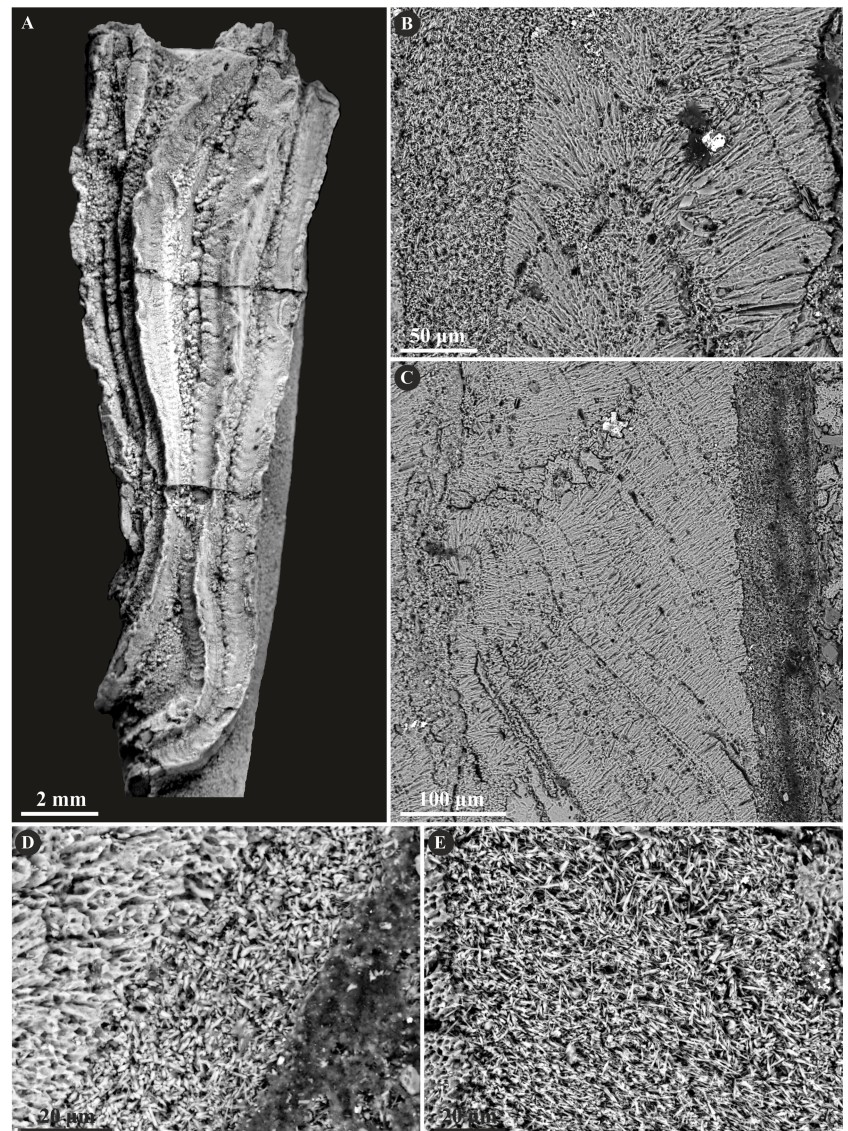

**Figure 11  Selected specimens of Serpulidae sp. from the Jurassic of Poland.** (A) Dense aggregation of closely spaced Serpulidae sp. 3 (see *Słowiński et al., 2022*) encrusting a fragment of a belemnite rostrum from the middle Bathonian of Gnaszyn Dolny (GIUS 8-3730/28). (B–E) Longitudinal sections of the Serpulidae sp. tube from the middle Bathonian of Gnaszyn Dolny showing two distinct layers: the external simple prismatic structure (SP) and internal irregularly oriented prismatic structure (IOP) (GIUS 8-3730/28).

species (*Vinn, 2007*). IOP microstructure most often builds the wall of single-layered tubes, or alternatively the inner part of multi-layered tubes (see *Vinn & Furrer, 2008*).

The representatives of Filograninae are generally characterized in the fossil record by more or less uniform fabrics of tube microstructures forming irregularly oriented structures, notably irregularly oriented prismatic structure (IOP). Secretion of such microstructures seems to be governed by the lower biological control of an animal, compared to more

advanced microstructures. The first confirmed appearance of such ultrastructure dates back to the Early Jurassic, possibly being present already in the Triassic (*Vinn, Jäger & Kirsimäe, 2008*). Such a simpler way of tube formation may indicate its plesiomorphic character and earlier origin (see also *Vinn, 2013*; *Ippolitov & Rzhavsky, 2015a*; *Ippolitov & Rzhavsky, 2015b*).

Another representative of this clade in the investigated material is the genus *Filogranula* comprising the recently discovered new species *F. spongiophila* (*Słowiński et al., 2022*), which has been studied with respect to ultrastructure for the first time, and by *F. runcinata*. The tube wall of *F. spongiophila* is single-layered and has a simple prismatic structure (SP), whereas *F. runcinata* possesses one well-preserved layer composed of spherulitic prismatic structure (SPHP) and a dubious IOP layer consisting of somewhat distorted, occasionally occurring areas of tiny, irregularly oriented crystals. As a result, the tube wall is interpreted as single-layered, composed of SPHP structure. Such microstructure occurs commonly in the clade Serpulinae comprising both formerly established "A" clades. The systematic position of the fossil *Filogranula* seems to be complicated, as the recent *Filogranula* is considered to be settled within Filograninae, relatively closely related to *Vermiliopsis* and *Metavermilia* (*Kupriyanova, ten Hove & Rouse, 2023*). While the Cretaceous *Filogranula cincta* (*Goldfuss, 1831*) is classified as belonging to Filograninae too, the Jurassic *Filogranula runcinata* was considered to be placed in Serpulinae, within the tribe Ficopomatini (see *Ippolitov et al., 2014*; *Koči & Jäger, 2015*), due to its prismatic microstructure, characteristic of the serpulid tubes appearing transparent in the recent species, such as *e.g.*, *Placostegus*. On the other hand, prismatic structures are not a fully satisfactory character in differentiating between former A and B clades. *Ippolitov & Rzhavsky (2014)*, *Ippolitov & Rzhavsky (2015a)* and *Ippolitov & Rzhavsky (2015b)* found out that some spirorbins may also have their tubes composed of regularly oriented prismatic microstructures, in spite of the fact, that Spirorbinae is phylogenetically closely related to Filograninae, which commonly form irregularly oriented structures.

Serpulinae constitutes two well-supported tribes–Serpulini and Ficopomatini (*Kupriyanova, ten Hove & Rouse, 2023*). The first unequivocal members of Serpulinae very likely existed since the Triassic (*Assmann, 1937*; see *Ippolitov et al., 2014*) if not even since the Permian (*Sanfilippo et al., 2017*; *Sanfilippo et al., 2018*). In contrast to Filograninae, which have mostly non-oriented structures, serpulids within this lineage have a more advanced biomineralization system and are capable of forming complex microstructures, such as *e.g.*, oriented, lamello-fibrillar, or oriented-fibrillar (see *Vinn & Furrer, 2008*; *Vinn & Kupriyanova, 2011*). During the Jurassic, serpulid microstructure diversity was relatively modest (ten types; see *Schlögl et al., 2018*; *Vinn, 2020*) compared to an array of contemporary and Cenozoic ones (*Vinn, 2007*; *Koči, Milàn & Jäger, 2023*). The most sophisticated microstructures in serpulids evolved in the Eocene (*e.g.*, (*Vinn, 2008*; *Buckman, 2020*; *Koči, Goedert & Buckeridge, 2022*). Although such complex microstructures as lamello-fibrillar or regularly ridged prismatic appeared quite late in the evolution of serpulids, the oriented prismatic structures evolved at the latest during the Middle Jurassic. Simple and spherulitic prismatic microstructures are very common and characteristic of the Jurassic Serpulinae and are considered to be apomorphic.

Oriented prismatic structures are unknown from any genus of Filograninae; however, they were found to persist in the clade Spirorbinae (*Ippolitov & Rzhavsky, 2015a*; *Ippolitov & Rzhavsky, 2015b*), which appeared in the Cretaceous. It may mean, that either it is plesiomorphic, or the prismatic structures have evolved at least twice.

The tribe Serpulini is represented in the Jurassic by characteristically coiled *Spiraserpula* and related *Cementula*, which is present in the herein material. *C. spirolinites* and *C. radwanskae* are both single-layered and possess their tube walls built of simple prismatic microstructure (SP), though in the latter it is somewhat obliterated. Simple and spherulitic prismatic microstructures exhibit the predominantly uniform orientation of calcium carbonate crystals which determines the optical transparency of the tubes (*Ippolitov & Rzhavsky, 2008*; *Zibrowius & ten Hove, 1987*), supported also by the dense arrangement of crystals and their large size (*Vinn & Kupriyanova, 2011*). In contrast, the recent tubes with IOP microstructure most often are optically opaque (*Vinn et al., 2008*).

All the main members of the Jurassic representatives of Ficopomatini are present in the investigated material. They contain genera with various morphotypes comprising a robust, single-keeled *Propomatoceros*, a three-keeled *Mucroserpula*, quadrangular tubes attributed most often to *Nogrobs*, and a well-defined, planispirally coiled *Placostegus*. Serpulids in this tribe commonly possess two layers of tube walls having the outer dense layer built of ordered prismatic structures (simple or spherulitic), and the internal layer composed of irregularly oriented prismatic structure. Two-layered tubes appeared in the Jurassic (*Vinn & Furrer, 2008*) and have their external layer denser than the internal, which is composed of thinner mineral microstructure. Dense outer protective layers (DOL) have been found in tube walls of serpulids inhabiting diverse environments (see *Vinn & Kupriyanova, 2011*). Consequently, rather than being dependent on the environment, the advent of DOLs during the Jurassic appears to be a significant evolutionary adaptation of serpulids. Except for some members of Spirorbinae (*Ippolitov & Rzhavsky, 2014*; *Ippolitov & Rzhavsky, 2015a*; *Ippolitov & Rzhavsky, 2015b*) and one species in the clade Filograninae, dense, outer layers exist exclusively in the clade Serpulinae (*Vinn & Kupriyanova, 2011*).

*Propomatoceros lumbricalis* and *Mucroserpula* possess here two well-distinguishable layers: internal IOP and external SPHP. A single *Propomatoceros* specimen may have three layers, which, however, is ambiguous, as its internal layer may be diagenetically obliterated, and therefore all the specimens are considered two-layered. As discussed previously (*Ippolitov et al., 2014*; *Słowiński et al., 2022*), it cannot be ruled out, that *Mucroserpula* and *Propomatoceros* may represent the same genus. The diagnostic characters of both genera are highly transitional, subjective, and dependent to a large extent on different variables, such as *e.g.*, ontogenetic stage (see *Słowiński et al., 2022*). Nonetheless, the microstructural fabrics of the two genera in our investigation are essentially the same and typical of Ficopomatini.

*Nogrobs* consists here of three species: *N.* aff. *quadrilatera*, *N?* aff. *tricristata*, and *N.* aff. *tetragona*. All of them are two-layered composed of internal IOP and external SP microstructure. Apart from *Nogrobs*, Jurassic quadrangular fossils of serpulids that share morphological similarities with this genus were attributed by various authors to different genera, partially of questionable validity—*Tetraserpula* (*Parsch, 1956*), *Tetraditrupa* (*Regenhardt, 1961*), *Glandifera* (*Regenhardt, 1961*), *Tubulostium* (*Stoliczka, 1868*),

*Tectorotularia* (*Regenhardt, 1961*), and *Ditrupula* (*Brünnich Nielsen, 1931*). *Kupriyanova & Ippolitov (2015)* examined and reviewed a number of microstructures of extant and fossil taxa having tusk-shaped, tetragonal in cross-section tubes, concluding, that these morphologically similar forms belong to several different genera, most likely being an effect of convergence. Not surprisingly, these recent taxa cannot be synonymized with the fossil *Nogrobs*. Nevertheless, the authors claim that at least the majority of fossil *Nogrobs* species may be members of a single clade, as opposed to morphologically-related recent species. According to its external tube layer composed of simple prismatic microstructure (responding to a transparent tube), the three investigated species fit well Ficopomatini, confirming previous ultrastructural studies (*Kupriyanova & Ippolitov, 2015*). It has to be noted, however, that the recent *Nogrobs grimaldii* (*Fauvel, 1909*) has been found to possess an opaque tube (*Kupriyanova & Nishi, 2011*).

The only exception not having a two-layered tube wall within this tribe is *Placostegus*, represented here by *P. planorbiformis*, which has a single layer made entirely of a simple prismatic structure (SP). Similarly, contemporary *P. tridentatus* (*Fabricius, 1779*) possesses a tube wall composed of a simple prismatic microstructure resulting in a completely transparent tube (*ten Hove & Kupriyanova, 2009*: 8, fig. 1F; *Vinn & Kupriyanova, 2011*).

The undetermined serpulid taxon Serpulidae sp., described recently by *Słowiński et al. (2022)* has been studied here with respect to its microstructure. Although the investigated specimens externally resemble a few genera, such as *Propomatoceros*, *Placostegus*, and *Metavermilia*, they do not exactly fit any of those taxa (see *Słowiński et al., 2022*). The specimens possess a tube wall composed of two layers—external simple prismatic (SP), and internal irregularly oriented prismatic microstructure (IOP), which is of a very close resemblance to other members of Ficopomatini, notably *Nogrobs*.

## Ecological implications and comparisons with sabellids

Serpulids perform biologically controlled biomineralization where their cellular activity regulates the nucleation and extracellular growth of the calcium carbonate crystals by the ion uptake from the surrounding water (see *Neff, 1971a*; *Neff, 1971b*) using a secretory epithelium, mediated and controlled by the organic matrix (*Vinn, Kirsimäe & ten Hove, 2009*; *Vinn, 2021a*). As a result, serpulids accomplish a specified crystal orientation within the tube wall, which is reflected by their variety of ultrastructural fabrics—the growth direction of crystals may be anisotropic, semi-oriented, or oriented (*Weedon, 1994*; *Vinn et al., 2008*). Apart from a matrix-mediated crystallization (*Vinn, 2021a*), an alternative explanation has been proposed recently to define the crystal orientation mechanism, which is based on a variable application of the serpulid's collar rotational force between the formation of ordered and unordered microstructures (*Buckman & Harries, 2020*). Considering the fact that the same growth increments may occur across zones with diverse ultrastructures makes (*Vinn, 2021a*; *Vinn, 2021b*) this model strongly disputable.

The serpulid biomineralization system differs from that of other tube-dwelling polychaetes. By the formation of the cylindrical and parabolic layers (see *Jäger, 1983*) serpulids are capable of forming multi-layered tubes arranged in distinct microstructures (see *Vinn et al., 2008*). Importantly, the parabolic layer is formed by adding secretory

increments to the rim of the worm's aperture allowing it to actively modify the external morphology according to the tube's sculpture, but also depending on the temporary ecological requirements. Such a solution results in a wide array of serpulid tube characters including attachment structures and base widenings of the tube, which combined with the ability to form several layers greatly improves the durability of the tube. Fossil cirratulids (*Vinn, 2009*; *Taylor et al., 2010*) and sabellids (*Vinn, ten Hove & Mutvei, 2008*; *Słowiński, Banasik & Vinn, 2023*) possess single-layered tube wall composed of a spherulitic prismatic structure. The presence of only cylindrical layer in sabellids strongly impairs their biomineralization abilities. Unlike serpulids having mostly chevron-shaped growth lines (see *Weedon, 1994*), they form their tubes by secreting calcareous material along distinct growth lamellae oriented parallel to the tube wall (*Vinn, ten Hove & Mutvei, 2008*; *Słowiński, Banasik & Vinn, 2023*). Subsequent increments are added to the internal surface of the tube, and therefore *Glomerula* is unable to modify its simple tube architecture. A much longer secretion zone in sabellids compared to serpulids enables them for a fast calcification incurring comparably lower physiological costs, which allows them to considerably prolong their tubes. On the other hand, such fast dispersal may be required by the diminishing inside of the tube, insufficient for the growing worm. Interestingly, cirratulids perform a double-phased, combined controlled, and influenced biomineralization. The products of an influenced biomineralization such as agglutinated xenolithic granules within a calcareous matrix, as well as the anisotropic orientation of different-sized crystals, indicate, that cirratulids govern even a weaker than sabellids control over biomineralization (*Guido et al., 2024*).

A much more advanced biomineralization system of serpulids and the resulting variety of microstructures are an important evolutionary adaptation for this taxon, which is an obligatory tube-dweller. The emergence of multi-layered tubes with dense, outer protective layers during the Jurassic could have been triggered by the intensified predation during the Mesozoic Marine Revolution (*Vermeij, 1977*). Interestingly, the development of dense, outer protective layers in spirorbins was shown to have a relationship with the type of paleoenvironment, its energy, and substrate kind (see *Ippolitov & Rzhavsky, 2015a*; *Ippolitov & Rzhavsky, 2015b*). With the skeleton development, serpulids could perform competitive strategies against other organisms instead of avoiding competition or temporary paleoenvironmental vagaries. Gradual increase in abundance of serpulids during the Mesozoic and Cenozoic (see *Ippolitov et al., 2014* for a review), and their advantage over other organisms allowed for higher plasticity and divergence of different morphotypes (*Vinn et al., 2024*). Different morphogenetic programs in turn enhanced their functional utilization not only by mechanical strengthening of the tubes but also by optimizing their space by planispiral coiling, providing a higher feeding tier, avoiding being overgrown or coated by sediment thanks to upward growth. Free-living serpulids with tusk-shaped tubes were presumably adapted to live in or on the surface of soft sediment during episodes of increased sedimentation rates (see *Vinn et al., 2024* for a review). Sabellid biomineralization system allowed them to quickly elongate their tubes but these polychaetes did not exhibit such diverse morphotypes. Instead, they could grow irregularly away from the initial point of encrustation and perform an opportunistic, fugitive strategy (see *Taylor, 2016*; *Słowiński,*

*Banasik & Vinn, 2023*). Additionally, it could be the outcome of biomineralization's lack of significance in this group as calcareous sabellids were restricted to a single genus in the family.

## CONCLUSIONS

The first thorough assessment of the ultrastructural diversity of Middle and Late Jurassic serpulid tubes has been conducted. The obtained data reveal a characteristic of the Jurassic serpulids, a relatively low diversity of ultrastructural fabrics, which generally correspond to certain clades recognized among extant taxa. Amongst 12 taxa representing two (Filograninae and Serpulinae) of the three main serpulid clades, six of them possess tube walls composed of a single layer, and six are two-layered. There are certain evolutionary trends in tube ultrastructures. The representatives of the clade Filograninae are single-layered and have their tube walls built of a primitive, irregularly oriented prismatic microstructure (IOP). The majority of members of possibly apomorphic clade Serpulinae possess two-layered tube walls, where the denser, external layer is composed of oriented, prismatic microstructures (either spherulitic (SPHP) or simple (SP)), and the internal is irregularly oriented prismatic (IOP). The exceptions are *Placostegus planorbiformis* and the genus *Cementula*, which are single-layered, and built of simple prismatic structure (SP).

Serpulid tube ultrastructures reflect their biomineralization abilities providing important paleoecological signatures. Formation of the regularly oriented microstructures of Serpulinae requires a higher biological control over biomineralization compared to the more primitive, anisotropic microstructures of Filograninae. The development of serpulid ultrastructure diversity was likely triggered by the evolutionary importance of the tubes for this group. A variety of microstructure types and the ability to form multi-layered tubes allowed serpulids to employ different morphogenetic programs, which had an impact on their functional utilization.

The differences in the biomineralization system between serpulids and other tube-dwelling polychaetes resulted from the importance of tubes for the former taxon. The complex biomineralization system of serpulids resulted in a multiplicity of forms and an ability to form robust, strongly ornamented tubes, which mechanically strengthened their durability. Although higher energy expenditure of skeletal secretion decreased the rate of tube formation, the solid attachment and skeleton robustness allowed for a competitive advantage over other encrusters. Conversely, sabellid primitive biomineralization abilities presumably were elicited by the unimportance of skeleton for this taxon resulting in a simplicity of forms. On the other hand, it enabled a fast spreading over the substrate, shunning competition and disadvantageous conditions by utilizing an opportunistic strategy of quick tube elongation.

## ACKNOWLEDGEMENTS

We would like to warmly thank Marian Külaviir (University of Tartu) and Arkadiusz Krzątała (University of Silesia) for their assistance with SEM. Tomáš Kočí and Alexei

Ippolitov, the journal referees, are greatly acknowledged for providing valuable comments and suggestions, which improved the final version of the manuscript.

### Funding
The authors received no funding for this work.

### Competing Interests
The authors declare there are no competing interests.

### Author Contributions
- Jakub Słowiński conceived and designed the experiments, performed the experiments, analyzed the data, prepared figures and/or tables, authored or reviewed drafts of the article, and approved the final draft.
- Olev Vinn conceived and designed the experiments, performed the experiments, authored or reviewed drafts of the article, and approved the final draft.
- Michał Zatoń conceived and designed the experiments, authored or reviewed drafts of the article, and approved the final draft.

### Data Availability
The SEM photographs of the investigated ultrastructures are available in the figures.

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
