# Peer review of "Ultrastructure of the Jurassic serpulid tubes–phylogenetic and paleoecological implications"

_PeerJ, doi:10.7717/peerj.17389_

## Round 0.1 · original submission · Minor Revisions

Please, address all the reviewers' comments either directly in the text or as a separate answer.

·

Basic reporting

.

Experimental design

.

Validity of the findings

.

Additional comments

The review
The tittle: Ultrastructure of the Jurassic serpulid tubes – phylogenetic and paleoecological implications
Authors: Jakub Słowiński1, Olev Vinn2, Michał Zatoń1
I congratulate to all authors on the beautiful new data about the ultrastructure of the Jurassic serpulids. I suggest minor revisions only. I suggest accepting this new study which contributed new data for knowledge of microstructures and ultrastructures of tube walls of serpulids. All my suggestions I have written in bubbles on the right side of the manuscript and in revision on the text for better work of colleagues after revisions. All Figures are prepared carefully and all pictures are sharp and focused. I recommend to accept and to print as soon as possible.
Dr. Tomáš Kočí PhD. 10. March, 2024 in Prague Letňany, Ivančická 581, 19900, Czechia.

All is marked in revision on textfor better work after revision.

I am much greatly appreciated for nice, beautiful reading of this new study. This hard work bring new datra for the knovledge of micro and ultrastructures of serpulid tubes.

·

Basic reporting

The paper is well-written, concise, and clearly explains what and why was done. It also has a good, thoroughly checked English language and clearly organised sections.

One remark considering the general structure of the paper - as this paper is based on the materials previously paleontologically described by the authors (Slowinski et al., 2022), I would offer to remove the discussion of the geological details from the text (section ‘Material and its provenance’), extending Table 1 instead (adding additional graphs ‘Age’, ‘Locality’, ‘Reference to the taxon description’, maybe ‘Reference to the locality description’). The details of geology: (1) are not important at all in the context of the present paper, (2) duplicate, sometimes almost literally, what was already published in Slowinski et al. (2022); (3) overload the reference list with many refs, which look a bit excessive for this paper.

Also, there are some inaccuracies about using the terminology, which require double-checking. For example, I am not sure that the term ‘growth lamellae’ was used by authors in the correct meaning (71-72; 230-231), i.e. in accordance with its original definition by Weedon (1994). Please also check the meaning of the word ‘parable’ (408) in the English language; I guess, particularly in this case, the authors meant ‘parabolic(al) layer’, which is the synonym of ‘chevron-shaped growth lamellae’. The using of the term ‘crystallization axis’ (171) also needs to be checked.

Some minor corrections, which are intended to improve the overall readability and to correct the representation of other authors' ideas in the discussed literature, as well as missing references to previous studies, are added as comments in the attached PDF.

Experimental design

There is one important question, considering the methods of investigation – the authors state that ‘taphonomically altered tubes have been discarded from further studies’ (95-96), but how particularly did they recognize them, what were the criteria? Plus, what is the difference between the discarded ‘taphonomically altered’ and the ‘diagenetically altered’ that were studied and discussed (199, 208, 215, 357)?
Please clarify this for the readers or rewrite the corresponding fragments.

Validity of the findings

The paper contains the statement that '...The emergence of multi-layered tubes with dense, outer protective layers during the Jurassic could have been triggered by the intensified predation during the Mesozoic Marine Revolution’ (433-435), which is also included in the abstract (27-30). However, this statement is not supported by serious discussion; thus, it looks declarative. I offer to remove it at least from the abstract. To note, the development of dense outer layers/structures in some members of the subfamily Spirorbinae (Ippolitov, Rzhavsky, 2015a,b and some additional papers referenced therein) was shown to be clearly connected with the habitat type (i.e. with ecological adaptations to the abiotic factors), not with the predation.
I also offer to remove the words ‘paleoecological implications’ from the title (even if the authors decide to leave the statement about the possible influence of the ‘Mesozoic marine revolution’) – because, in reality, you do not have them. The discussion around the general advantage of serpulid tubes vs other polychaete tubes is more about macroevolutionary strategies/adaptations than about ‘palaeoecology’. The same remark is valid about the section titled ‘Ecological implications and comparisons with sabellids’ – this is more about macroevolution than about ecology.

Additional comments

1) Taxa names in the text are sometimes cited with the full initials of their authors (166-167, 168, 190-191; 218-219 etc). This should be done only in two cases: (A) if two or more palaeontologists had the same family name (e.g., J. Sowerby and J. de C. Sowerby) or (B) if two or more authors with the same family name are cited (where one can be a taxonomist, while the other is a geologist, for example). But not in other cases. Therefore, initials for A. Goldfuss, H. Regenhardt, etc., should be removed.
2) there are some taxonomic names introduced without the authors (369) – please either add authors of these taxa or cite several papers where these taxa are used.
3) The authors should be more accurate when they compile the reference list. In the case of PeerJ, the editors take responsibility for editing the list, according to the policy of the Journal. However, in general, you should follow the ‘writing hygiene’ yourself and the order of papers when you have multiple references to papers with the same first author should be the following (automatically generated bibliographies typically do not consider this!):
- individual papers by the author, arranged by year
- papers by the author with one co-author, arranged by the co-author’s name and then by the year (if there are several papers by the same pair of co-authors).
- papers by 3+ authors, arranged by year (without consideration of the second co-author’s names).

---

## Round 0.2 · accepted · Accept

Thank you for a thorough review and taking into account all the comments. I am happy to accept the manuscript for publication.

·

Basic reporting

I congratulate for Jakub Slowinski et al for very beautiful resubmited paper..
I recommend to publish asap. Authors improved this manuscript as very carefuly. e. g. references were improved, all figures were prepared and created as top level=excellently, all figures ar with sharp focus and all are visible and readable. The structrure of this resubmited article is clear, readable etc.
Tom Kočí

Experimental design

This excellent. This is primary basic research of nice field of ultrastructure and microstructure of tube wall of serpulids. All reasearch data, question, comparison are well defined, relevant and meaningfull. This paper contribute new data of microstructure of jurassic serpulds. This paper is writen in top level, with high technical and ethical standard. Methods were described as well and indetails. In overall this paper is writen ,prepared in top hight level.
Tomáš Kočí

Validity of the findings

Thi new resubmited version if much more better that submitted version.
I see key data for study of serpulids from Jurassic deposits - authors prepared
microstructures and ultrastructures of serpulid tube wall, this new data
as base for following reasearch on this nice filed.
Their validity and conclusion are excellent.
Tom Kočí

Additional comments

I recommend accept in this form and try to publish asap. This resubmitted paper bring new light of microstructures of jurassic serpulid.
Congratulations for Jakub Slowinski team, excellent paper. I am looking forward to read in printed version.
Tom Kočí

·

Basic reporting

I thank the authors for elaborating my remarks and suggestions on the first version of the manuscript in a constructive but reasonably critical manner - it is always pleasant to see such a way of author-reviewer interaction.

The only thing that still needs to be corrected is the word 'parable' (612, 614).
Although you refer to the book by Jaeger (1983), his original term in German is 'Parabelschicht', which should be translated into English as 'parabolic layer', not 'parable layer'. The word 'parable' in the English language means a 'short story, especially in the Bible, that shows you how you should behave' (literal definition from the Cambridge Dictionary). Please correct this.

Experimental design

no comment

Validity of the findings

no comment

Additional comments

no comment